# Nursing Students' Preferences for Learning Medical and Bioscience Subjects: A Qualitative Study

**Lars Kyte** [1,*] , **Ingrid Lindaas** [2] , **Hellen Dahl** [2] , **Irene Valaker** [1] , **Ole T. Kleiven** [1] and **Solveig Sægrov** [1]

1    Department of Health and Caring Sciences, Faculty of Health and Social Sciences, Western Norway University of Applied Sciences, Campus Førde, Svanehaugvegen 1, 6812 Førde, Norway; irene.valaker@hvl.no (I.V.); ole.kleiven@hvl.no (O.T.K.); solveig.nelly.segrov@hvl.no (S.S.)

2    Department of Health and Caring Sciences, Faculty of Health and Social Sciences, Western Norway University of Applied Sciences, Campus Haugesund, Bjørnsonsgate 45, 5528 Haugesund, Norway; ingrid.lindaas@hvl.no (I.L.); hellen.dahl@hvl.no (H.D.)

\*    Correspondence: lars.kyte@hvl.no

**Abstract:** (1) Background: There are considerable challenges and concerns related to learning medical and bioscience subjects (MBS) in nursing education and integrating this knowledge into nursing. The aim of this study was to explore what learning methods nursing students prefer when studying MBS, and how this learning may be enhanced to facilitate the integration of these subjects into nursing. (2) Methods: Individual interviews with 10 nursing students. Transcripts from the interviews were analysed by systematic text condensation and the COREQ checklist for qualitative studies was completed. (3) Results: Students prefer varied and active learning methods in MBS. The participants in the study highlighted both organised tutorials in groups and working with fellow students outside of organised teaching. All participants used educational videos. Learning MBS by drawing was appreciated both during lectures and in student-initiated colloquia. Strategies that favour in-depth learning were appreciated, and it was found that lectures did not have to cover the entire curriculum. Teachers' attitudes toward students also were seen to have a considerable impact on students' motivation for learning. (4) Conclusion: Applying active learning methods and focusing on the most relevant topics in MBS appears to improve students' ability to integrate this knowledge into nursing; teachers should also be aware of their role as a motivator.

**Keywords:** in-depth interviews; learning methods; medical and bioscience subjects; nursing education; qualitative method

## 1. Introduction

Medical and bioscience subjects (MBS) are regarded as an important base for practising nursing and providing safe patient care [1–3]. Knowledge from MBS is necessary to ensure that the decisions nurses make in practice maximise the health impact and avoid harm to patients [4]. Furthermore, the ability to integrate theories from MBS when practising nursing correlates with improved patient outcomes [5]. In Norwegian nursing education, MBS include anatomy, physiology, biochemistry, microbiology, pathophysiology, and pharmacology. Many of these subjects are taught early in the nursing education programme. This forms a knowledge base that students need to apply in clinical practice later in their studies [6].

There are considerable challenges and concerns related to learning MBS in nursing education and integrating this knowledge into nursing practice [5,7]. Challenges include the large curriculum content, large class sizes, structuring MBS in the curriculum, course organisation and learning environment, student motivation and concentration, the integration of bioscience in clinical practice, and high failure rates at exams [3,7–9].

Many new nursing students are anxious about studying bioscience, and there is a need for improved support to facilitate their learning [1]. Students' prior competence in

mathematics, chemistry, and biology is related to their achievements in bioscience [9] and pre-entry qualifications in science for nursing students have been proposed [7]. However, measures regarding entry qualifications will not solve the problem alone, and the focus must also be on higher education institutions and their responsibility to promote an organised learning environment that enhances learning in these subjects [7].

It can be a major challenge to teach MBS in a way that suits students of different ages and with varying learning styles [10]. Moreover, different studies show diverging results regarding what learning methods are preferable in facilitating students' learning process. One study showed that most students learned best kinaesthetically and that social learning was beneficial [11]. In a recent study, students following a flipped classroom model were both more satisfied and obtained better exam results than students following traditional lectures. This model was particularly effective for low-achieving students [12]. Another study found that most nursing students regarded classroom lectures as the most efficient approach when teaching anatomy and physiology. Group seminars were felt to be a waste of time for one-third of the students, while small group tutorials contributed well to the learning process [10]. However, although many students appreciate small group learning, this does not suit everyone, and there are challenges regarding time spent and tutors' competence [13]. A literature review revealed that there is a lack of knowledge on how to best support students' learning in bioscience subjects. New ways of teaching using digital tools, simulation, and different kinds of games enhance students' engagement and increase student satisfaction. However, the students' satisfaction with the teaching offered does not appear to correlate with their exam results [8].

The challenge regarding MBS in the nursing curriculum is also more complicated and involves more than simply considering students' achievement in exams in these subjects. Exam results probably do not tell the whole story about the benefits of this knowledge, as the reason for nursing students learning MBS is not to acquire this knowledge itself, but to apply this knowledge in nursing. Successful integration of MBS theory in nursing practice may be of great importance for patients; the appropriate application of bioscience in practice is vital in order to be able to offer safe and competent patient care [4,5].

This is the second article from a study exploring different aspects regarding the integration of MBS in nursing. The purpose of the first article was to describe how students experience the integration of medical and bioscience knowledge in nursing education. The findings presented in that article showed that students highlighted the importance of linking theory to practice [6]. Based on this, the aim of this study is to explore what learning methods nursing students prefer when studying MBS, and how the learning may be enhanced to facilitate the integration of these subjects into nursing.

## 2. Materials and Methods

### 2.1. Design

In this study, we applied a qualitative approach. The chosen design follows a hermeneutic-phenomenological tradition, as the aim was to explore students' own experiences and thoughts and capture their opinions about MBS in nursing education [14,15]. To achieve this, an interview guide for individual in-depth interviews with open-ended questions was constructed. The interview guide covered different aspects related to MBS in nursing education. The interview guide was partly semi-structured, as the interviewer was given the option to ask supplementary questions if necessary.

### 2.2. Recruitment of Participants

A total of 10 nursing students, two male and eight female, in the second or third year of their undergraduate studies, from two campuses at a university college in Norway, participated in this study (Table 1).

**Table 1.** Study participants.

| Campus | Participant and Gender (f/m) |
|---|---|
| A | 1 f |
| | 2 f |
| | 3 m |
| | 4 m |
| | 5 f |
| B | 1 f |
| | 2 f |
| | 3 f |
| | 4 f |
| | 5 f |

None of the participants dropped out of the study. A criterium for inclusion in the study was that the participants had completed their clinical placements in both surgical and medical wards in a hospital. The reason for this criterium was that during these placements students have to use medical and bioscience knowledge acquired earlier in their studies. To obtain permission to recruit participants for the study, we contacted the administration of the faculty responsible for nursing education at the students' university college. Students were subsequently recruited through information about the study in class. Of those who registered as participants, five students were randomly selected from each campus to participate in the study.

*2.3. Data Collection*

The COREQ checklist for qualitative studies [16] was completed. The research team consisted of four registered nurses (RN), one biologist, and one medical doctor (MD). At the time of the study, all researchers were employed as associate professors. All researchers (four women and two men) had experience from earlier research. Some of the researchers had been lecturers to some of the participants, but they had no personal relationship with them.

All interviews were performed face-to-face during 2019 and lasted approximately 30–60 min. The interviews took place on the campuses where the participants were studying. All participants on both campuses provided extensive information about how they perceived learning bioscience in their education, and how this knowledge base was integrated into nursing practice during their studies. When the data collection was completed, interviewers noted that saturation was obtained. Each interview was conducted by one of the researchers (I.V., I.L., or H.D.), audio-recorded, and subsequently transcribed. Some field notes were also made in connection with the interviews.

*2.4. Data Analysis*

After reading through the transcripts to gain an overall impression, all researchers discussed the content and agreed on the major themes of the interviews [14,17,18]. These themes covered content relevant to both this article and the first article in the project. The themes relevant to this article were chosen, and further analysis of this article was performed in accordance with the description of systematic text condensation by Malterud (2012, 2017) [14,18]. This part of the analysis was mainly performed by three of the researchers. The chosen themes formed the basis for code groups. Then, the meaning units from the transcripts were sorted into these code groups. Code groups were divided into subgroups, and during the analysis process code groups evolved and changed, in accordance with the description of the method [14,18]. Table 2 shows the code groups

and subgroups that evolved through the analysis process with examples of meaning units corresponding to the different codes.

**Table 2.** Code groups and subgroups with examples of corresponding meaning units.

| Examples of Meaning Units | Subgroups | Code Groups |
|---|---|---|
| "There has been a good variation in everything. Mostly, it has been teaching, but the fact that one . . . draws, shows drawings, some videos and then explanation again" (B5)<br>"The seminars, I think these were useful, because then you have to kind of use both hemispheres of the brain" (A2) | Use of diverse learning methods | Active and varied learning methods |
| " . . . we actively listened to the others, we discussed and changed a little bit and discussed, so it was certainly helpful." (A3) | Collaborative learning | |
| "Yes, because if they manage to link the material to stories and their own experiences . . . , then we get a little more insight into what awaits us in working life" (B1) | Learning by understanding connections | In-depth learning |
| " . . . lecturers wanted to say as much as possible in the shortest possible time" (B1) | Knowledge must be manageable | |
| " . . . it's about showing commitment, . . . entering a classroom and wanting to teach [students] something" (B5)<br>"A good lecturer for me is someone who doesn't rush through what we're going to learn in the curriculum, but actually takes time if there's something you don't understand to go through it a bit more, and include the students, not just stand and lecture, but actively actually ask." (B4) | During organised learning activities | The lecturer as a motivator for students' learning |
| "the most important thing when it comes to the lecturer is that you have someone who cares . . . and say hello when you meet them in the hallway. It makes it much more personal, especially when it becomes more personal for my own learning" (A1) | Outside of organised learning activities | |

In the next step of the analysis, the meaning units of each subgroup were reduced to condensates. These condensates consisted of content from several meaning units given by different participants and were written as first-person statements ("artificial quotes"), according to the method [14,18]. Then, an analytic text for each subgroup was synthesized based on these first-person statements and quotes from the interviews. This analytic text formed the basis for the presentation of the results. No software was used in the analysis.

*2.5. Ethical Considerations*

The study was approved by the Data Protection Officer for Research at the Norwegian Centre for Research Data (NSD, no 287137). All participants were given written information about the project and signed consent forms prior to participation. To protect anonymity, participants' names were not used either in audio recordings or in transcripts. The recorded data and transcripts from the interviews were saved on a protected data server for research purposes belonging to the Western Norway University of Applied Sciences.

## 3. Results

The major themes revealed by the analysis were active and varied learning methods, in-depth learning, and the teacher as a motivator for students' learning. This corresponds to the code groups presented in Table 2.

*3.1. Active and Varied Learning Methods*

Many of the participants emphasised the importance of applying varied teaching methods during MBS lectures. It was stated that using PowerPoint alone is too monotonous, and that visual resources such as images, films, or animations can make it easier to understand and remember the material. Some highlighted "Kahoot!", a student response system (SRS), which encourages student activity by allowing students to answer questions in class

by using a device. Several also pointed out that drawing during lectures enables better learning. It was emphasised that when students draw together with the lecturer, it makes it easier to understand the connections being taught in the subject. One of the participants put it this way:

> *If you sit and are taught [by the lecturer] all the time, then it becomes boring, but if you draw a bit or gets out some sheets of paper . . . , you learn with all your senses, so it is about using different methods* (B5).

Several participants said that they benefited greatly from digital resources as a supplement to other learning methods. All participants used educational videos, and several emphasised that this is useful, both because one can watch the videos when one has free time and because one can watch them several times in order to understand something better.

> *But it's when it is the physiological . . . what happens in the lungs, what happens in the kidneys, why the heart works that way, the chemical stuff. Then you may need to have it explained in different ways . . . it needs to mature, so you may have to listen to it several times* (B1).

It is also helpful to draw and take notes when hearing and watching videos because then you learn by using motor skills and several senses at the same time.

Several participants highlighted the benefits of seminars and other learning methods that required students to be active themselves and "do something". "And you remember in another way when you do it yourself than if you are only told it in a lecture" (A2). An example of this, which some of the participants remembered "very well", was running up and down a staircase while breathing through straws, in connection with learning about nursing for patients with chronic obstructive pulmonary disorders (COPD). Other examples were the use of anatomy models where students could pick up and look at the organs, dissection of the heart and kidneys (from pigs), and the cultivation of bacteria (from the hands). "It's nice to have visual resources like that; they're useful—it creates interest. We get to do it ourselves and can see with our own eyes . . . what is actually in the body" (A3).

The participants in the study appreciated organised tutorials in groups in MBS, where students can ask and students can answer, supervised by the teacher. In smaller groups the threshold for asking is lower; this makes you talk and discuss together. " . . . it brings out that discussion, that conversation, and you can learn in a different way . . . " (A1).

Many of the participants also preferred to work with fellow students outside of organised teaching. Quizzes, drawing on boards, and videos were used as learning methods in student-initiated colloquia. Some also pointed out that by explaining something to others, you remember things better: "I acquire a lot of knowledge by talking and explaining . . . and if you can explain it to someone, you've understood it so well that you know it" (A1).

However, most people do not want to work in groups all the time, and some students expressed a greater need to work on their own than others:

> *I prefer to work alone when I am reading. I think it's very nice to have groups, and to sit and discuss, but when I'm really going to work, I have to be alone. I can't concentrate if there's noise. So we work in different ways* (B1).

Some mentioned challenges regarding group work, as some group members contribute more than others. This may be frustrating when the group has to deliver a joint product.

*3.2. In-Depth Learning*

It was argued that lecturers prioritising the most important elements within each topic and not rushing through everything in the curriculum during the lectures may have an impact on students' sense of mastery.

> *. . . all the advanced diagnoses, we do not have to go through them during lectures. We can read about that ourselves . . . it is much better to know a lot about a little than a little*

*about a lot when it comes to lectures. Then I think we could have left the classroom with a greater sense of mastery . . .* (A4).

Discussing patient cases in class when learning MBS was also suggested, as discussion provides opportunities to hear other people's thoughts on a problem. It was argued that hearing others may widen the horizon and have an impact on clinical thinking: " . . . if you hear how other students think, and how they come up with a solution, it widens the horizon a little bit, and I think you eventually get a broader clinical view from hearing others" (B3).

One of the participants particularly highlighted the importance of not delaying reading until just before the exam, as that forces students to learn a lot in a short time. " . . . to some extent, when you cram something, you can achieve a good grade, but that way you don't retain it" (A1). It was argued that you need to understand something to be able to simplify it when explaining it to patients or relatives (in clinical practice later in the studies), i.e., if you cram everything, you are only able to reproduce what you have crammed.

Simulation-based learning was also highlighted as a learning method that enhanced students' understanding of clinical practice; as one of the students put it: " . . . in the simulation we have the opportunity to make mistakes and learn from them . . . " (A3).

### 3.3. The Teacher as a Motivator for Students' Learning

Regardless of the choice of learning methods, it was emphasised that teachers' attitudes towards students are important for students' learning and that it means a lot for students' motivation that the teacher cares about them. Several participants explained how teachers' ways of meeting students, both in organised learning activities and elsewhere, influenced learning.

In organised learning activities, it was emphasised that a good lecturer is someone who knows the subject matter, has good pedagogical abilities, shows commitment, and wants to teach students something. Several mentioned the importance of the lecturer including the students and creating dialogue in the classroom by both questioning the class and creating a safe framework that allows students to ask questions. " . . . he uses the board and is very engaged, and we have learned a lot from him . . . and that's also because he was so good at having dialogue with the class" (A5).

Several pointed out that what teachers do outside of organised learning activities is also important. The teacher's availability both at college and via e-mail was highlighted, and it was expressed how important it is for learning that the teacher cares about the students and treats them as individuals. It becomes more personal, even for students' learning, if the teachers say hello to students when they meet them in the hallway. It is also motivating when students find that the results of exams mean something not only to students but also to the teachers. One participant described what it means for motivation when the teacher cares, outside organised learning activities as well:

*There was one thing we absolutely did not understand, and then we went to talk to the lecturer, and then he spent an hour getting our group of five people to understand this before the exam . . . it's the lecturers who do such things who make me want to do well too* (B4).

## 4. Discussion

The aim of this study was to explore what learning methods nursing students prefer when studying MBS, and how that learning may be enhanced to facilitate the integration of these subjects into nursing.

### 4.1. Encourage Active Learning

Participants highly appreciated learning activities in MBS that forced them to be active learners. The term "active learning" is defined in several ways, and the definition has evolved over time, but it often means learning through activities or discussions that make students reflect on their understanding, by keeping them both mentally and physically

active [19]. Freeman et al., 2014 constructed the following consensus definition: "Active learning engages students in the process of learning through activities and/or discussions in class, as opposed to passively listening to an expert. It emphasizes higher-order thinking and often involves group work" [20] (pp. 8413–8414). Following this definition, traditional lectures with a lecturer speaking to an audience are not viewed as active learning, and achieving active learning for students in lectures is challenging [21].

### 4.2. Variation during Lectures Is Needed

Lectures were widely used at both campuses, and the participants had various experiences regarding how lectures were delivered. A key result, highlighted by all participants, was the need for variation and student activity during lectures. In line with the definition of active learning [20] and the recommendations from Biggs (1999) [21], participants wanted lecturers to encourage students to participate in dialogue and discussions, something that may be challenging in large classes. They also highlighted the importance of creating a safe learning environment where students dare to speak up.

Two ways of promoting activity in class that were appreciated by participants in the study were the use of drawing and the use of "Kahoot!", a student response system (SRS). Several studies support the benefit of drawing, especially in anatomy [22–24]. Drawing makes lectures less boring, as students have to work actively [23]. When students draw simultaneously with the lecturer, it requires them to watch, reflect upon what they see, and draw themselves. This combination of visual stimuli, cognitive reflection, and motor activity may contribute to the learning process, and, in our study, it was mentioned that drawing enhanced the understanding of connections. This is in accordance with Fernandes et al., 2018 who proposed that drawing has an impact on a memory trace consisting of elaborative, motoric, and pictorial components [25]. There is also evidence that drawing may help students better retain knowledge [22,24], something that is important in being able to apply knowledge from MBS when practicing nursing later in their education.

"Kahoot!", an SRS that can deliver questions to students in a quiz format, was mentioned in our study as something that promotes variation. A large meta-analysis showed that "Kahoot!" has a positive effect on students' motivation and learning, and improves teacher-student interaction [26]. However, the extent to which it helps students retain information may vary [27]. On the other hand, this kind of tool may be used to promote discussions in class on topics in biology [28].

Nevertheless, the potential for discussion and dialogue in large classes is often limited, and learning methods other than lectures may be more suitable for achieving this, as several participants in our study clearly indicated. Learning methods that forced students to "do something" were appreciated, and various examples of these kinds of learning methods were given; many students learn best kinaesthetically and benefit from social learning [11].

### 4.3. Learning Together

Participants in our study appreciated the group tutorials led by teachers. They also highlighted the benefit of discussing bioscience with peer students in groups organised by the students themselves. According to Freeman (2014), active learning often involves group work [20]. Working together in groups also includes more than discussing topics verbally. Participants in our study used quizzes, watched online resources together, and made drawings together. Thus, working together implies a variety of learning methods, something that may help students to understand connections in the subject [10].

Data from a survey based on a national exam in anatomy, physiology, and biochemistry for nursing students showed that students working in groups obtained significantly higher grades than students who did not work together in groups [9]. When students discuss subjects they do not understand with fellow students, it motivates further learning [29]. Some of the participants in our study expressed that explaining difficult subjects to others strongly enhanced their own learning and facilitated the retention of knowledge. If students better retain knowledge from MBS, it will probably promote the integration

of this knowledge into their nursing practice later in their education. When students try to explain something to fellow students, it generates valuable learning and also makes them aware of what they need to learn more about [29]. The combination of preparing to teach and explaining to others enhances learning [30]. The benefit of explaining to others was mentioned in our study in regard to the context of learning difficult material, and collaborative learning seems to be especially important when tasks are complex [31]. When students work together, they not only learn the subject studied but also experience that they need each other to achieve their goals [31]. Biggs (1999) argues that learning is a way of interacting with others, changing our conceptions so that we see something in a different way [21]. This is what one of the participants in our study highlighted when she stated that you can get a broader clinical view by listening to others.

However, participants also highlighted challenges regarding group work, for instance when the group has to deliver a joint product. This indicates that preparing students for group work and what it demands is important. Peer assessment of other group members' contributions has been found to enhance students' engagement in group work and make them take responsibility for their learning [32]. However, students learn in various ways, and their different perceptions regarding group work must be accepted. Teachers and fellow students may contribute to creating a positive learning atmosphere for students who feel insecure when working in groups [33]. Trust among group members is vital in order for a group to function, as students who rely on each other demonstrate stronger cognitive engagement and contribute more actively to group work [34]. Working together as students may also be advantageous regarding future work, as most nurses work in teams either in hospitals or in primary health care.

### 4.4. Videos Should Be Combined with Other Learning Methods

Most participants in our study highly appreciated the use of teaching videos when studying MBS. A meta-analysis covering more than 100 trials from various areas of higher education revealed that the use of videos led to improved learning outcomes [35]. Participants in our study particularly appreciated the opportunity to watch videos on difficult topics over and over again, something that may reduce cognitive overload [35].

Most participants preferred to use videos as a supplement to lectures or other learning methods. A recent report showed no clear connection between students' exam results and their use of an online video resource in bioscience, used by most students pursuing nursing education in Norway [9]. Several studies, including two major meta-analyses, support the idea that videos work best when combined with learning in face-to-face classes. This kind of blended approach also seems to be more efficient than classroom teaching alone [9,35,36]. Therefore, when the students in our study expressed that videos enhanced learning in MBS, it may be due to the effect of combining this kind of resource with other learning methods, such as lectures and discussions with peers.

### 4.5. Promoting In-Depth-Learning May Facilitate the Use of MBS Theory in Nursing Practice

Nursing students perceive bioscience as important for providing safe patient care [2,3], and it is essential for students to learn the fundamental principles of bioscience during their first year in order to facilitate the use of this knowledge later in their studies [7]. When teachers succeed in demonstrating the clinical relevance of bioscience for nursing, it motivates students to learn and may contribute both to a better understanding of the subjects taught and better exam results [9,37]. If teaching MBS focuses solely on theoretical aspects without linking to clinical practice, it may be reasonable to ask to what extent students' grades in these subjects reflect the usefulness of this kind of knowledge in clinical practice.

Our study highlights that rushing through all the curriculum during lectures may reduce students' experience of coping. Coping is important to promote students' interest and motivation for learning, thereby promoting deeper learning [38]. When students perceive an overload of tasks, it is demotivating for their learning [39]. This implies that

teachers in MBS should help students to focus on the important and most relevant topics in the subject, rather than every detail in the curriculum, in order to better prepare them for the complexity of patient care. In patient care, knowledge from MBS must be used in connection with other kinds of knowledge such as nursing skills, ethics, and social sciences. This is far more complex than answering exam questions on anatomy and physiology during the first year of a nursing degree. This complexity should be made clear to students during their studies [3]. Simulation-based learning (SBE) may be a way of introducing this complexity [40]. In our study, SBE was highlighted as a safe way of training. When using SBE, different kinds of knowledge must be combined; this makes SBE an appropriate area for in-depth learning.

*4.6. A Supportive Learning Environment*

Several participants in the study clearly stated that teachers' concern for their students and their engagement and involvement in the student learning process are important for students' motivation. This is not primarily about the teacher giving a brilliant performance in class, but rather about the teacher's values and how these values can create an engaging learning environment that supports students [41]. The importance of this should not be underestimated, as the teacher is central to the students' learning environment. A supportive learning environment is important when students are to learn the basic principles of bioscience, and if this fails it may reduce their understanding of the importance of bioscience later in their studies [7]. This may counteract the integration of MBS in nursing practice. A recent systematic review showed that integrating bioscience in nursing by using clinical scenarios in the first year of the education program was associated with an increased understanding of bioscience and that supporting students with this integration is needed [42].

Nursing students are demotivated by negative comments from the teacher, or if they feel that the teacher is not interested in the students' learning [39]. A study among students in various kinds of higher education showed that the teachers' involvement in the student learning process and students' feeling of competence have an impact on students' achievement [43]. Thus, teachers' attitudes towards students are of great importance for students' motivation and learning. This is important throughout the education programme, and probably no less important at the beginning of the programme when students are not familiar with the demands and routines of higher education. As MBS is usually taught early in the nursing curriculum, teachers in these subjects should be aware of the importance of caring and involvement in students' learning process for students' motivation.

**5. Study Limitations**

One limitation of this study is the size of the study, as it was conducted at just two different campuses. Although students at these campuses had experienced several different learning methods, the results may not be directly transferable to other institutions with other educational approaches and learning methods. In addition, the students who registered as participants in the study may have been particularly interested in the topic and are not necessarily representative of all students.

Another limitation is the gender distribution of the participants, as only two of the participants were men. However, this reflects the gender distribution in Norwegian nursing education.

The interviews were conducted by three different researchers. Although all interviews followed the same interview guide, there may have been differences in how the interviews were carried out, and to what extent supplementary questions were asked during interviews.

**6. Conclusions**

In order to provide high-quality care for patients, nursing students need to learn MBS and be able to integrate this knowledge into nursing. This requires that education

programmes focus on learning methods that support in-depth learning in these subjects. The knowledge must be manageable, and students need to understand how MBS theory relates to nursing practice. To achieve this, teachers must help students to focus on the most relevant topics in the subjects to better prepare them for the integration of this knowledge into patient care.

Nursing students prefer varied learning methods that actively engage them when studying MBS. These include drawing, watching videos, and teachers interacting with students during lectures. In addition, students appreciate working together and discussing, actively listening to others, and explaining professional issues to fellow students. This enhances learning and facilitates the retention of knowledge, thereby supporting the integration of this knowledge into their nursing practice later in their education.

A good teacher-student relationship has a considerable impact on students' motivation for learning. In class, the lecturer must involve students and contribute to a safe and supportive learning environment that allows students to ask questions. What teachers do outside of organised learning activities is also important, and students are motivated when they perceive that their teachers care about them and their learning process.

This study did not look at students' grades. Further studies are needed to investigate whether students' grades in MBS are related to their ability to retain this knowledge and integrate it into clinical practice.

**Author Contributions:** Conceptualisation: S.S., L.K., I.L., H.D., I.V and O.T.K. Formal analysis: L.K., I.L., H.D., I.V., O.T.K. and S.S. Investigation: I.V., I.L. and H.D. Methodology: L.K., I.L., H.D., I.V., O.T.K. and S.S. Project administration: S.S. Supervision: S.S. and L.K. Writing—original draft: L.K. Writing—review and editing: L.K., I.L., H.D., I.V., O.T.K. and S.S. All authors have read and agreed to the published version of the manuscript.

**Funding:** This research received no external funding.

**Institutional Review Board Statement:** The study was conducted according to the guidelines of the Declaration of Helsinki and approved by the Data Protection Officer for Research at the Norwegian Centre for Research Data (NSD, no 287137).

**Informed Consent Statement:** Written informed consent was obtained from all subjects involved in the study.

**Data Availability Statement:** The data presented in this study are not publicly available due to ethical reasons. The participant consent and approval from the Norwegian Centre for Research Data (NSD, no 287137) prevents us from sharing data in public repositories. However, the data will be available from the corresponding author upon reasonable request. Transcripts from the interviews are saved on a protected data server for research purposes belonging to the Western Norway University of Applied Sciences. According to the informed consent signed by the participants, the data will be deleted two years after the project was finished (the project was finished on 31 July 2022).

**Acknowledgments:** We are grateful to the students who participated in this study.

**Conflicts of Interest:** The authors declare no conflict of interest.

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
