# Peer review of "Nursing Students’ Preferences for Learning Medical and Bioscience Subjects: A Qualitative Study"

_nursrep, doi:10.3390/nursrep13020055_

Round 1

Reviewer 1 Report

Thank you for the opportunity to review the manuscript. Overall, a current topic for a broader readership and further exploration of this topic is certainly unique, especially to explore to explore what learning methods nursing students prefer when studying MBS, and how the learning may be enhanced to facilitate the integration of these subjects into nursing in Norway.

A few questions / comments and suggestions:

In Line 138-160, any COREQ to ensure the rigor of this research study, relevant to the study is not clear.

In Line 292-294, relevant to the study is not clear.

In Line 310-311, how to promote deeper learning, must elaborate more clear, relevant to the study is not clear.

In Line 320-322, clearly elaborate “the retention of knowledge”, relevant to the study is not clear.

In Line 344-347, elaborate “other kinds of knowledge, relevant to the study is not clear.

In Line 414, more elaboration for reduction of the understanding of the importance of bioscience, relevant to the study is not clear.

Reviewer 2 Report

First of all, I want to congratulate the authors of this study. In fact, it is a relevant topic for the academic and scientific community.

The manuscript has a good structure, and according the metodological criterion.

It is important to write possible suggestions for future studies.

Please see the comply with referencing standards (in the topic of bibliographic references).

Reviewer 3 Report

I would like to congratulate the authors for the study they present!

1. Brief Summary

The authors prudently conducted a research study on an interesting subject. They explored what learning methods nursing students prefer when studying MBS, and how the learning may be enhanced to facilitate the integration of these subjects into nursing. They main findings showed that applying active learning methods, focusing on the most relevant topics and having a teacher with a motivator attitude can contribute to improve the students’ ability to integrate the MBS into nursing.

2. Comments

In my opinion, the study is well designed and scientifically sound, the methods are appropriate, the results are well presented and described, and the results are appropriately discussed.

I only have a few comments/suggestions that I think they can contribute for the improvement of the manuscript.

#1 On line 140, references are written in red. I suggest you change them to black.

#2 Table 2: This table should be removed from the "Data Analysis" subchapter and moved to the "Results" chapter after line 172. Furthermore, I suggest that the column "Example of meaning units" be removed, since the authors are giving examples throughout the description of the results.

#3 In the subchapter "Data Analysis" the authors should mention whether or not they used any software for the data analysis. If so, they should identify which software was used.

#4 On line 278, the authors wrote “The aim of this study is …”. I suggest they change it to ““The aim of this study was …”, because the "Discussion" chapter should be written in the past tense.

#5 Authors must update the references, as around 53% are more than 5 years old.

Kind regards
